# Optimal Tracking Control for Nonstrict-Feedback Nonlinear Systems

1st Yuanbo Su
*Navigation College*
*Dalian Maritime University*
Dalian, China
yuanbosu2019@163.com

2nd Qihe Shan
*Navigation College*
*Dalian Maritime University*
Dalian, China
shanqihe@163.com

3rd Fei Teng
*College of Marine Electrical Engineering*
*Dalian Maritime University*
Dalian, China
brenda_teng@163.com

4th Tieshan Li
*School of Automation Engineering*
*University of Electronic Science and Technology of China*
Chengdu, China
tieshanli@126.com

*Abstract*—This paper investigates the optimized tracking control problem for nonlinear single-input-single-output systems in nonstrict-feedback form. An intelligent approximation-based optimized backstepping control strategy is developed using reinforcement learning. An improved variable separation method is presented to address the algebraic loop issue arising from the implementation of virtual optimized controls. The designed actual optimized controller ensures that the system's output converge to a small neighborhood of the reference signal and all the signals in closed-loop systems are semiglobally ultimately uniformly bounded.

*Index Terms*—tracking control, nonlinear systems, nonstrict-feedback form, optimized backstepping

## I. Motivation

Optimized backstepping consensus control for nonlinear systems has been widely investigated owing to its engineering applications. The goal of optimized control is to design reinforcement learning-based controller which makes system's output to agree on a desired signal, and realizes a tradeoff between the control performance and the control cost. Nevertheless, the considered system models are the strict-feedback form, which can be expressed as

$$\dot{X}_i = X_{i+1} + F_i(\bar{X}_i) \tag{1}$$

$$\dot{X}_n = U + F_n(X_n) \tag{2}$$

$$Y = X_1 \tag{3}$$

where, for $i = 1, 2, ..., n$, $\bar{X}_i = [X_1, \ldots, X_i]^T$, $U$ and $Y$ are the state vector, the control input, the output of the system, respectively. $F_i(\cdot)$ is an unknown nonlinear function. Nonstrict-feedback dynamics are often needed in practice

such as robot systems and electrical systems. As for the reinforcement learning-based optimized backstepping control of nonlinear systems in nonstrict-feedback form, the proposed virtual optimized local controls can include whole states of controlled systems since the nonlinearity term $F_i(\cdot)$ can be the ones with all states, which can be expressed as $F_i(X_i)$, where $X_i = [X_1, \ldots, X_n]^T$. This will lead to the problem of algebraic loop, which motivates our work.

## II. Method

To design the optimal controller for nonstrict-feedback nonlinear systems, where system's dynamic includes the nonlinear functions of whole state variables, the algebraic loop problem must be solved. An improved variable separation technique is proposed by establishing a new variable separation inequality and utilizing the structure feature of fuzzy logic systems. Specifically, the designed virtual optimal controllers for systems are independent of the whole state variables in this paper.

## III. Conclusion

This paper has developed an adaptive optimal tracking control method for nonstrict-feedback nonlinear systems. A new variable separation method has been proposed to address the problem of algebraic loop in the implementation of virtual optimal controllers. It can be proven that the designed actual optimized controller ensures that the systems output converge to a small neighborhood of the reference signal and all the signals in closed-loop systems are semiglobally ultimately uniformly bounded.

This work was supported in part by the National Natural Science Foundation of China under Grants 52201407, 51939001, 52371360, 61976033, and 62173172; in part by the Natural Foundation Guidance Plan Project of Liaoning under Grant 2019-ZD-0151; in part by the Fundamental Research Funds for the Central Universities under Grant 3132019345; and in part by the Liaoning Revitalization Talents Program under Grant XLYC1908018.
