# OpenReview forum: "Optimal Tracking Control for Nonstrict-Feedback Nonlinear Systems"
_IEEE.org/ICIST/2024/Conference — IEEE ICIST 2024 Conference Submission_

### Official Review · Reviewer_tfrM · 2024-08-21
**Lack of innovation**

**Rating:** 2
**Confidence:** 5

**Review:**

This article lacks innovation. And the writing structure of the article is not clear.

---

### Official Review · Reviewer_2Rdv · 2024-08-22
**reject**

**Rating:** 2
**Confidence:** 5

**Review:**

This paper investigates the optimized tracking control problem for nonlinear single-input-single-output systems in nonstrict-feedback form. Unfortunately, we regret to inform you that due to insufficient innovation in your paper, it cannot be published at this conference.

---

### Official Review · Reviewer_GJrp · 2024-08-26
**The author is advised to optimise this paper**

**Rating:** 6
**Confidence:** 4

**Review:**

This paper proposes an intelligent approximation-based optimized backstepping control strategy based on  reinforcement learning and presents a variable separation method. To address the problem of algebraic loop in the implementation of virtual optimal controllers, there is a new variable separation method. It is recommended that the authors give a quantitative analysis.

---

### Decision · Program_Chairs · 2024-09-08

Reject